# Upconversion nanocomposite for programming combination cancer therapy by precise control of microscopic temperature

Xingjun Zhu[1], Jiachang Li[1], Xiaochen Qiu[1], Yi Liu[1], Wei Feng[1] & Fuyou Li[1]

Combinational administration of chemotherapy (CT) and photothermal therapy (PTT) has been widely used to treat cancer. However, the scheduling of CT and PTT and how it will affect the therapeutic efficacy has not been thoroughly investigated. The challenge is to realize the sequence control of these two therapeutic modes. Herein, we design a temperature sensitive upconversion nanocomposite for CT-PTT combination therapy. By monitoring the microscopic temperature of the nanocomposite with upconversion luminescence, photothermal effect can be adjusted to achieve thermally triggered combination therapy with a sequence of CT, followed by PTT. We find that CT administered before PTT results in better therapeutic effect than other administration sequences when the dosages of chemodrug and heat are kept at the same level. This work proposes a programmed method to arrange the process of combination cancer therapy, which takes full advantage of each therapeutic mode and contributes to the development of new cancer therapy strategies.

---

[1] Department of Chemistry & Institutes of Biomedical Sciences & State Key Laboratory of Molecular Engineering of Polymers, Fudan University, 220 Handan Road, Shanghai 200433, China. Correspondence and requests for materials should be addressed to W.F. (email: fengweifd@fudan.edu.cn) or to F.Y.L. (email: fyli@fudan.edu.cn)

Combination cancer therapy uses more than one treatment mode to treat patients, which exhibit additive killing effect to cancer to reduce progressing or relapse[1]. Nanomaterials become powerful tools in cancer treatment[2–6]. The concept of combination therapy was adopted recently to develop new anticancer nanoagents[7–9]. Nanosystems with multiple treatment modes involving chemotherapy, radiotherapy, photothermal therapy, photodynamic therapy, etc., have been reported so far[10–18]. Among them, the combination of chemotherapy (CT) and photothermal therapy (PTT) is quite common, and in most reported works, these two treatment modes were initiated at the same time[19–28]. Typically, a chemodrug is loaded in thermal-responsive compound and integrated with photothermal agent to form a CT-PTT therapeutic nanocomposite[21]. In this way, drug release is activated by photothermal effect. Considering that photothermal effect is usually monitored by macroscopic device (thermometer or infrared camera), the overall temperature (defined as apparent temperature) has to be kept at a high level to ensure drug release, which causes simultaneous administration of chemodrug and heat without adequate schedule. However, as evidenced by a number of studies, the sequence of CT and PTT will determine the final therapeutic efficacy[29–33]. For example, giving chemodrug (doxorubicin) prior to thermal therapy results in better antitumor effect than simultaneous intake[29, 31, 34]. If the sequence of the treatment modes can be properly arranged and the therapeutic efficacy improves, drug dosages may be reduced, thereby resulting in lesser side effects[29, 32, 34]. In our view, monitoring and controlling nanocomposite's temperature (defined as the eigen temperature) is the key to schedule the treatment modes in CT-PTT combination therapy. If eigen temperature can be precisely controlled to just initiate drug release without excessive heat leading to PTT, the separation of the two treatment modes will be realized.

To detect the eigen temperature of the nanocomposite, microscopic temperature sensing method is indispensable. Luminescent sensing technology emerges as a powerful tool for the detection of microscopic temperature, owing to its high sensitivity and the use of small size probes. To date, several types of optical temperature probes such as small molecular dyes, quantum dots, polymers, and lanthanide-doped upconversion nanophosphors (Ln-UCNPs) have been developed[35–42]. Thanks to the unique luminescence process of converting low-energy excitation into high-energy emission, Ln-UCNPs exhibit multiple merits for temperature sensing in physiological environment such as low detection limitations, absence of autofluorescence, and less photodamage[43–55]. Recently, Ln-UCNPs combined with photothermal agents were used to detect the eigen temperature during photothermal process with improved accuracy of PTT[56].

Herein, a temperature responsive upconversion nanosystem (TR-UCNS), which contains photothermal agent (Octabutoxyphthalocyanine palladium (II), abbreviated as PdPc) and thermal responsive drug release unit (1,2-dipalmitoyl-sn-glycero-3-phosphocholine, abbreviated as DPPC), is built to achieve programmed CT-PTT combination cancer therapy. Photothermal effect of TR-UCNS is generated by 730 nm laser irradiation. Eigen temperature during photothermal process is interpreted by measuring the temperature sensitive upconversion luminescence (UCL) of $Er^{3+}$ with the excitation of 980 nm laser. Power density of 730 nm laser, corresponding to photothermal effect, is tuned in light of UCL feedback to schedule CT and PTT sequentially (Fig. 1).

## Results

### Synthesis and characterizations of TR-UCNS.
The synthesis of TR-UCNS was achieved by a stepwise method, which is shown in

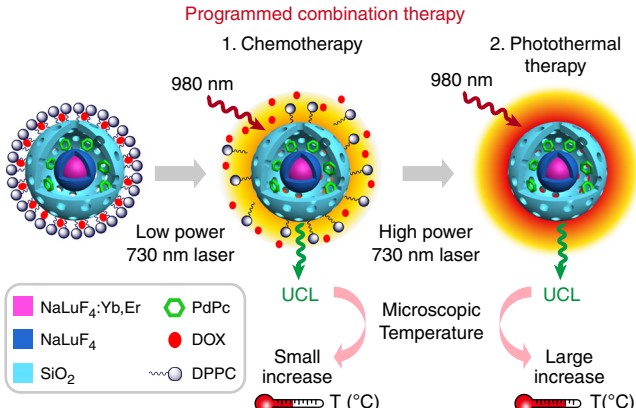

**Fig. 1** Schematic diagram of programmed combination therapy. Temperature responsive upconversion nanosystem (TR-UCNS), which contained photothermal agent (PdPc) and thermal responsive drug release unit (DPPC micelle), was built to perform programmed CT-PTT combination cancer therapy. By monitoring the microscopic temperature of TR-UCNS with upconversion luminescence (UCL), the power density of photothermal excitation source (730 nm laser) can be tuned accurately to initiate thermal-controlled drug release and photothermal therapy stepwise, thereby programming the combination therapy. Programmed combination therapy showed better therapeutic effect at a lower drug dosage than conventional combination therapy, from which the treatment modes were initiated simultaneously

Supplementary Figure 1. Firstly, the lanthanide-doped upconversion nanoparticles (NaLuF$_4$:20%Yb,2%Er, abbreviated as UCNPs) are prepared by a solvothermal method[57]. As shown in the transmission electron microscopy (TEM) image, the morphology of UCNPs were spherical with a size of 13.4 nm in diameter (Fig. 2a). To enhance the upconversion luminescence (UCL), an undoped NaLuF$_4$ layer was grown on the surface of UCNPs to form a core–shell structure (NaLuF$_4$:20%Yb,2% Er@NaLuF$_4$, here abbreviated as csUCNPs). The shape of csUCNPs remained spherical and the size increased to 20.5 nm in diameter (Fig. 2b). X-ray powder diffraction (XRD) patterns of csUCNPs indicated that both kinds of nanoparticles were the hexagonal phase of NaLuF$_4$ (Supplementary Figure 2). The obtained csUCNPs with oleate as ligands can be easily dispersed in non-polar solvents. To realize the loading of photothermal agent and chemodrug, a porous-silica hollow shell was constructed outside csUCNPs through a hard template etching process. Here, a modified reverse microemulsion method was employed to form the etchable multilayered silica template[58]. Two kinds of organosilicone precursors, tetraethyl orthosilane (TEOS) and N-[3-(trimethoxysilyl)propyl]ethylenediamine (TSD), were used to deposit the silica layers with an adding sequence of TEOS, TEOS and TSD, and then TEOS. The silica layers coated on csUCNPs exhibited a whole thickness of ~14 nm and kept the monodispersity of nanoparticles (Fig. 2c). As demonstrated in previous works[58, 59], the co-hydrolysis of TSD and TEOS makes the inner silica layer much looser than the outer layer, which is a hard template of the hollow cavity. With the treatment of 5% hydrofluoric acid solution, the template silica layer was etched and the outer silica layer was preserved to constitute the hollow shell. As shown in Fig. 2d, the yolk-shell-like nanoparticles (NaLuF$_4$:20%Yb,2%Er@NaLuF$_4$@Yolk-Shell SiO$_2$, abbreviated as YSUCNP) were uniform in morphology with a diameter of 48.1 nm and the silica shell was 5.8 nm in thickness. Nitrogen adsorption-isotherm analysis confirmed the existence of a porous structure in YSUCNP (Supplementary Figure 3a), with a

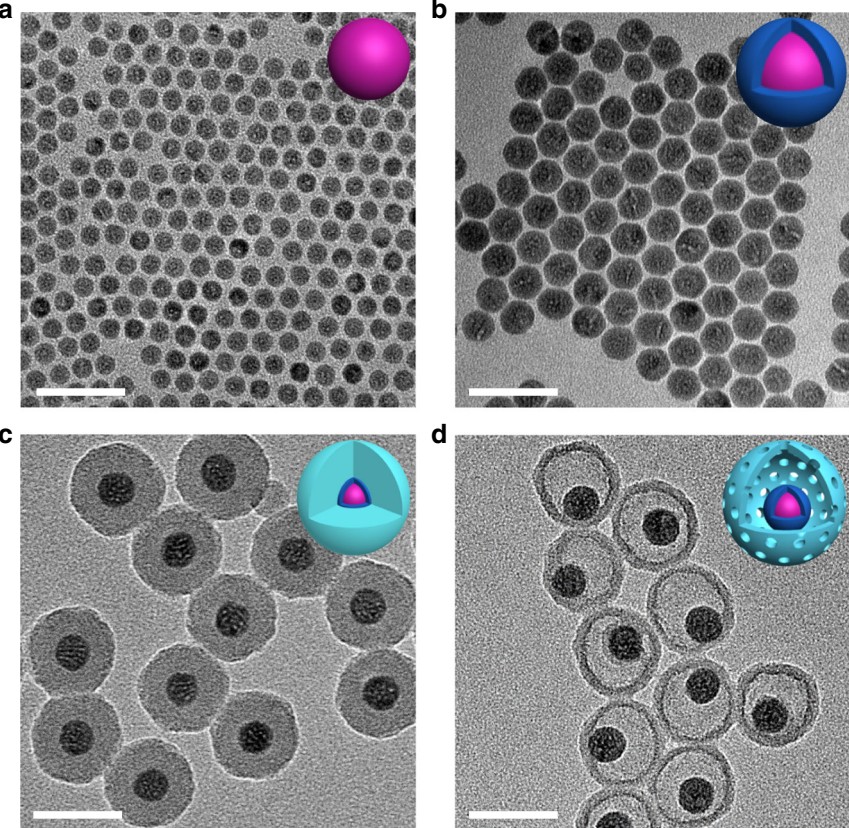

**Fig. 2** Morphology characterization of the nanocomposite. TEM images of **a** NaLuF$_4$:Yb,Er (UCNPs), **b** NaLuF$_4$:Yb,Er@NaLuF$_4$ (csUCNPs), **c** NaLuF$_4$:Yb,Er @NaLuF$_4$@SiO$_2$, and **d** NaLuF$_4$:Yb,Er@NaLuF$_4$@Yolk Shell-SiO$_2$ (YSUCNP). Scale bars were defined as 50 nm

pore volume of 0.296 cm$^3$ g$^{-1}$ and a Brunauer–Emmett–Teller (BET) surface area of 110 m$^2$ g$^{-1}$. The pore size distribution curve of YSUCNP had a sharp peak at 3.4 nm, which indicated the mesoporosity of the silica shell (Supplementary Figure 3b).

The mesoporous silica shell and hollow cavity of YSUCNP make it possible to load the functional guest molecules. PdPc with strong absorption at near-infrared region was loaded in YSUCNP to endow the nanoparticles with photothermal conversion ability. Fourier-transform infrared spectroscopy revealed the appearance of stretching bands of C−H bonds after PdPc loading (Supplementary Figure 3c). Energy-dispersive X-ray (EDX) mapping also confirmed the existence of palladium element in the nanoparticles (Supplementary Figure 4). This indicates the successful loading of PdPc in YSUCNP. To achieve the thermal-responsive chemodrug release, DPPC, of which the micelle can dissociate with the elevation of temperature, was selected as the vehicle of the chemodrug[60–62]. Doxorubicin (DOX), a commonly used chemodrug, was chosen as a typical example in this study[29]. The assembly of the drug release unit on the surface of YSUCNP-PdPc to form YSUCNP-PdPc@DPPC-DOX (TR-UCNS) was performed according to a previously reported method[63]. TEM image of TR-UCNS showed a new coating layer outside the silica shell (Supplementary Figure 5). According to the data shown in Supplementary Figure 3d and e, the molar of PdPc loaded in 20 mg YSUCNP is 1.28 μmol (2 mg × 76.4% × 1197.8 g mol$^{-1}$). The molar of 20 mg YSUCNP is 0.31 nmol, so the number of PdPc in each YSUCNP nanoparticle is calculated by 1.28 μmol/0.31 nmol. Hence, there are ~4129 PdPc molecules in each YSUCNP particle. DOX laden on 10 mg YSUCNP-PdPc (which contain 0.14 nmol YSUCNP nanoparticles) is 0.115 μmol (2.5 mM × 50 μl × 91.8%), so there are ~821 (calculated by 0.115 μmol/0.14 nmol) DOX molecules on each nanoparticle.

TR-UCNS exhibited a typical upconversion luminescence (UCL) emission of Er$^{3+}$ ions under 980 nm laser excitation (Fig. 3a). Although the luminescence underwent slight suppression after the loading of PdPc and the adjunction of drug delivery unit, the overall intensity of TR-UCNS was still sufficient to carry out UCL-based temperature sensing and bioimaging. The existence of PdPc in TR-UCNS brought about a considerable absorption band centered at ~750 nm, which can be utilized to perform photothermal therapy under near-infrared light excitation (Fig. 3b). To evaluate the photothermal properties of TR-UCNS, photothermal conversion efficiency was measured under the excitation of 730 nm laser. As shown in Supplementary Figure 6, the final heat-generation efficiency ($\eta$) was 54.2%, which was higher than the gold nanostructures (e.g., 21%), Cu$_9$S$_5$ (25.7%) and some organic materials[56]. Such high conversion efficiency is quite important, since the eigen temperature can be elevated to the level for drug release and PTT under more moderate laser irradiation.

**Drug release by using eigen temperature.** In the previous study, the eigen temperature of photothermal nanomaterials detected by UCL can be utilized to control the range of photothermal effect, thus improving the accuracy of PTT. Here we further applied the eigen temperature that just initiated the drug release without generating excessive amount of heat for restricting photothermal effect, thus separating CT and PTT in the combination therapy. In TR-UCNS, the upconversion core (NaLuF$_4$:20%Yb,2%Er) served as the reporter of eigen temperature during photothermal process. Since $^2H_{11/2} \rightarrow {}^4I_{15/2}$ and $^2S_{3/2} \rightarrow {}^4I_{15/2}$ transitions (UCL emissions centered at 525 and 545 nm, respectively) in Er$^{3+}$-doped upconversion system are in thermal equilibrium ruled

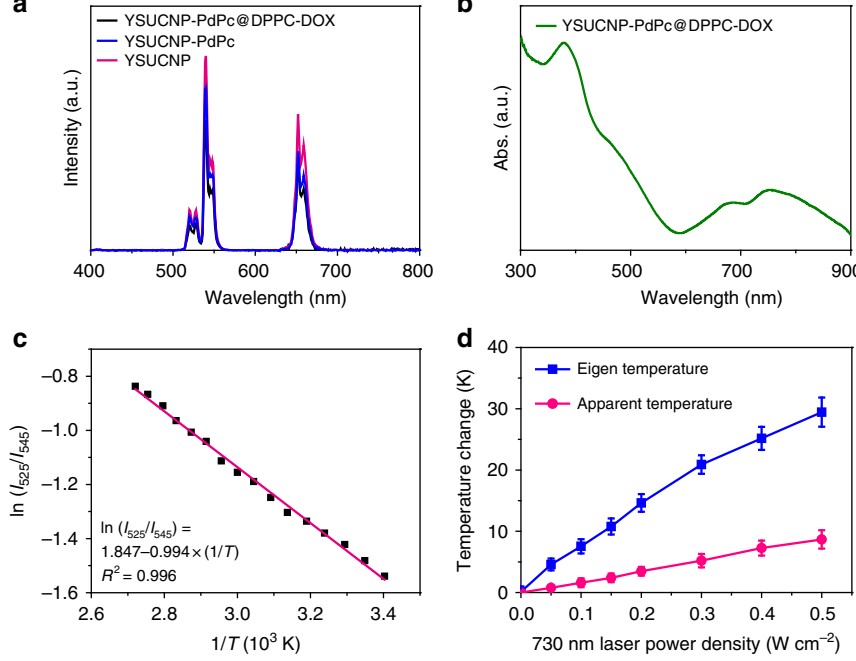

**Fig. 3** Optical and temperature sensing properties of TR-UCNS. **a** Upconversion luminescence spectra of YSUCNP, YSUCNP-PdPc, and YSUCNP-PdPc@DPPC-DOX (TR-UCNS) in the aqueous solution with the same concentration of $Er^{3+}$ ($2.5 \times 10^{-5}$ mol l$^{-1}$). The excitation wavelength of upconversion luminescence is 980 nm. **b** Absorption spectrum of TR-UCNS. **c** A plot of ln ($I_{525}/I_{545}$) versus $1/T$ to calibrate the thermometric scale for TR-UCNS. $I_{525}$ and $I_{545}$ indicate the intensities of UCL emission of the $^2H_{11/2} \rightarrow ^4I_{15/2}$ and $^4S_{3/2} \rightarrow ^4I_{15/2}$ transitions, respectively. **d** Elevation of apparent temperature (A.T.) and eigen temperature (E.T.) of TR-UCNS (0.5 mg ml$^{-1}$) in aqueous dispersion under irradiation with a 730 nm laser at various power densities. Average value of A.T. and E.T. under different time points were given based on three times measurements. Error bars were defined as s.d. ($n = 3$)

by the Boltzmann factor, temperature can be reflected by the intensity ratio of the two emission bands[45, 56]. The relationship between temperature and UCL emission conforms to the following equation (Eq. 1):

$$(I_{525})/(I_{545}) = C \exp(-\Delta E/kT), \quad (1)$$

where $I_{525}$ and $I_{545}$ are the integrated intensity of UCL emission derived from the $^2H_{11/2} \rightarrow ^4I_{15/2}$ and $^2S_{3/2} \rightarrow ^4I_{15/2}$ transitions, respectively; $C$ is a constant determined by degeneracy, spontaneous emission rate, and photon energies of the emitting states in the host materials; $\Delta E$ is the energy gap separating the two excited states; $k$ is the Boltzmann constant; $T$ is temperature using the Kelvin scale.

Firstly, UCL spectra of TR-UCNS aqueous solution at variable temperatures were collected to obtain a calibration curve for optical temperature sensing. As shown in Fig. 3c, the dependence of ln $(I_{525})/(I_{545})$ on the inverse of temperature $(1/T)$ followed a linear relationship, which could be fitted as $\ln(I_{525})/(I_{545}) = 1.847 - 994 \times (1/T)$ ($T$ given in K). Then, TR-UCNS solution at 37 °C was irradiated by 730 nm laser with different power densities to generate photothermal effect. UCL emission intensities at each point were detected to investigate the eigen temperature during the photothermal process. With the elevation of laser power density, the eigen temperature and apparent temperature of TR-UCNS rose correspondingly (Fig. 3d). The eigen temperature had a much greater magnitude of increase than apparent temperature, which was consistent with the previous results. It is worth noting that the eigen temperature of TR-UCNS reached to 41.6 °C (apparent temperature reached to 37.8 °C) when the power density of laser was at 50 mW cm$^{-2}$ (Fig. 3d). As reported in the previous study, the assembly consisting of DPPC molecules would completely dissociate at this temperature. We also observed the phenomenon that DOX release underwent a sharp improvement by heating TR-UCNS with an external device to such temperature (Fig. 4a). Hence, the DOX release process under 730 nm laser irradiation at 50 mW cm$^{-2}$ was further investigated to verify the feasibility of realizing drug release by using eigen temperature. As shown in Fig. 4b, nearly 95% of DOX released from TR-UCNS within 10 min under laser irradiation. On the other hand, the release rate was only 10.6% if TR-UCNS was externally heated to the same apparent temperature (37.8 °C) with laser irradiation at 50 mW cm$^{-2}$. These results indicate that eigen temperature plays a more direct role in drug release relying on photothermal effect, and drug release of TR-UCNS can be successfully achieved under quite moderate eigen temperature (41.6 °C). In consideration of the rather limited heat conduction of nanoparticles at such eigen temperature, the photothermal effect should be well confined to initiate CT only without triggering PTT.

**Programmed combination therapy in vitro.** To achieve the eigen temperature sensing in the cell, the calibration curve describing the relationship between temperature and UCL was measured in TR-UCNS-labeled MIA PaCa-2 (human pancreatic adenocarcinoma) cells. The curve exhibited a good linear behavior, similar to the one measured in the aqueous solution (Fig. 5a). Then, the 730 nm laser power densities applied to perform CT and PTT in cells were determined according to the calibration curve. It can be seen that the eigen temperature reached to about 41.5 °C, which was sufficient to initiate the drug release, under 46 mW cm$^{-2}$ 730 nm laser irradiation. Moreover, the eigen temperature was elevated to 45.4 °C, which can induce a certain degree of thermal damaging effect, under the power density of 140 mW cm$^{-2}$ (Fig. 5b). Meanwhile, the apparent temperature of the above two sets of laser power were limited within 39 °C (Fig. 5b).

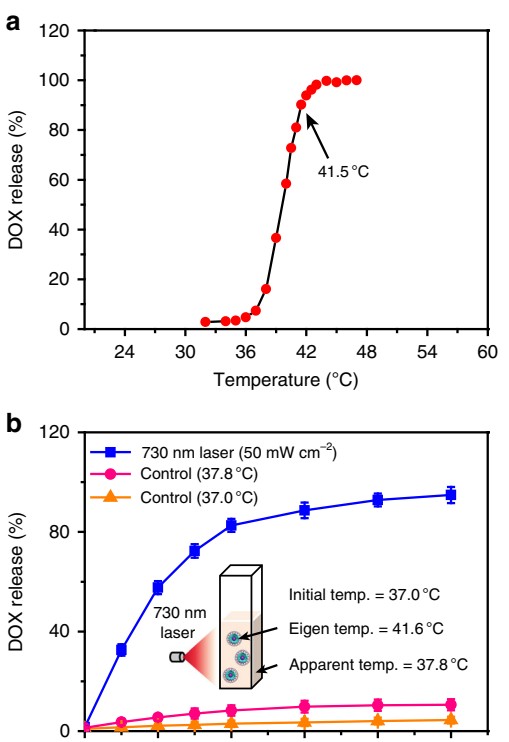

**Fig. 4** Drug release behavior of TR-UCNS. **a** DOX release behavior of TR-UCNS aqueous dispersion (0.5 mg ml$^{-1}$) under external heating. The temperature varied from 32 to 47 °C. The release rate was normalized to 47 °C. It should be noted that the release rate at 41.5 °C reached to ~90%. **b** DOX release rate of TR-UCNS aqueous dispersion (0.5 mg ml$^{-1}$) versus time upon 730 nm laser irradiation (50 mW cm$^{-2}$). Error bars were defined as s.d. ($n = 3$). The eigen temperature and apparent temperature of TR-UCNS reached 41.6 and 37.8 °C, respectively. In control experiments, the temperature of the solution was kept at 37 and 37.8 °C, and the release rate was much lower than that achieved by laser irradiation. These results indicate that eigen temperature during photothermal therapy is the key factor for drug release

To confirm the feasibility of separating CT and PTT with the help of eigen temperature, cell viability under different operations were evaluated. MIA PaCa-2 cells were used throughout the cell experiments. The design of programmed combination therapy is shown in Fig. 6a. MIA PaCa-2 cells were labeled with TR-UCNS and then imported into the process of programmed combination therapy. In this program-like process, the power densities of 730 nm laser were regulated by the eigen temperature of TR-UCNS to control the photothermal effect. Lower laser power density was used for drug release for chemotherapy (Command 1#), while the higher one was used for photothermal therapy (Command 2#). Programmed combination therapy was performed as a sequence of Command 1# to Command 2#. Time interval between Command 1# to Command 2# is 4 h, which is determined by the therapeutic efficacy in vitro (Supplementary Figure 8).

TR-UCNS or 730 nm laser alone did not cause obvious killing effect (Supplementary Figure 7d and e). When the cells incubated with TR-UCNS were irradiated by 730 nm with low power density (46 mW cm$^{-2}$) to initiate DOX release, a 20% decrease in viability was observed, compared with the group without irradiation. If the cells were treated in the same condition, but incubated with TR-UCNS in the absence of DOX, only a very slight decrease was presented in cell viability (Fig. 6b). These

results indicate that the decrease in cell viability under low power irradiation was mainly attributed to the killing effect of DOX releasing from TR-UCNS, rather than the heat. If the cells were irradiated with high power density (140 mW cm$^{-2}$), which not only initiated drug release for CT, but also gave excessive heat to induce PTT, the cell viability reduced to 51.3% (Fig. 6b). However, there was still a large portion of cells that survived. Although the enhancement of DOX dose or laser power density may help to eliminate the cells thoroughly (Supplementary Figure 7b), these practices will also bring about damages to normal tissues or even the whole body due to high concentrations of chemodrug and mass heat conduction. To avoid this issue, we controlled the sequence of CT and PTT by modulating the existing two sets of power densities (46 and 140 mW cm$^{-2}$, which is represented by 'L' and 'H', respectively) to verify whether the scheduling of treatment modes in combination therapy will improve the therapeutic effect. Four groups of cells were all irradiated by 730 nm laser two times and the irradiation power density of the four groups were L→L, H→H, L→H, and H→L, respectively. Interestingly, nearly all cells were killed (only 1.3% cells survived) when they were irradiated at 46 mW cm$^{-2}$ first and then at 140 mW cm$^{-2}$ (L→H) (Fig. 6). Such arrangement of irradiation power could initiate CT first and then PTT. As shown in Fig. 5c, when the cells were irradiated under 46 W cm$^{-2}$ for 10 min, eigen temperature mapping images illustrated a relatively small temperature elevation to about 41 °C, and the fluorescence signals of DOX were also detected in the cytoplasm. When the cells were irradiated under 140 W cm$^{-2}$ for 10 min, the eigen temperature raised to ~45 °C and the cells were dead as their nucleuses could be stained with propidium iodide (PI) (Fig. 5d). Other sets of irradiation power such as H→H and H→L, which initiated CT and PTT at the same time, and L→L, which initiated CT only, showed limited killing effects (Fig. 6). These results implied that, in CT-PTT combination therapy, the sequence of scheduling CT first, followed by PTT (L→H), had better therapeutic efficacy (about 39 fold enhancement of therapeutic effect) than the conventional operation in which CT and PTT were given simultaneously (H→L). It should be noted that the dosages of DOX in TR-UCNS (the concentration of DOX loaded in TR-UCNS is 2.5 μM in the incubation solution) and the heat used in programmed combination therapy were rather low and cannot meet the needs of conventional therapy (Supplementary Figure 7a and b). Furthermore, the sequencing of the treatment modes is realized by monitoring the eigen temperature, which is difficult for conventional strategies that depend on apparent temperature. Previous studies have shown that heat shock protein (HSP) generated from the preliminary thermal treatment will protect the cells from the harm induced by chemodrug[64–68]. To explore the possible reason for the enhancement of the therapeutic effect in programmed combination therapy compared to conventional approaches, we treated the cells with HSP70 activator (115-7c) first and then evaluated the cell viabilities. It turned out that the cells pre-treated with 115-7c survived in programmed combination therapy and the operations simulating programmed combination therapy with external heating and pure DOX treatment (Supplementary Figure 9). However, the cells without the treatment of 115-7c were effectively eliminated as expected. Moreover, western bolt analysis was used to detect the HSP expression to give more evidence for the possible mechanism of programmed combination therapy. β-Actin in the cells was used as internal control and HSP70 was used to represent the HSP. Anti-HSP70 antibody [C92F3A-5] (ab47455, Abcam) was used as primary antibody for HSP70 determination. Anti-β-Actin antibody (ab8226, Abcam) was used as the primary antibody for β-Actin determination. The results showed that when cells were treated at 41.5 °C for 5 min (the same

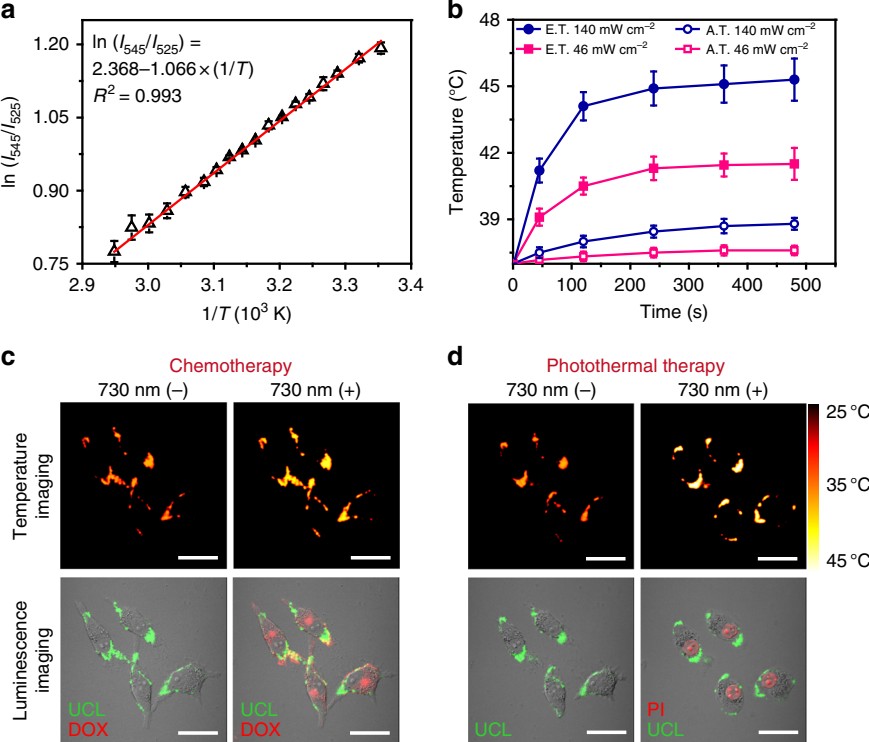

**Fig. 5** Eigen temperature sensing and drug release at cell level. **a** A plot of $\ln(I_{525}/I_{545})$ versus $1/T$ to calibrate the thermometric scale for TR-UCNS in MIA PaCa-2 cells. Average values of $I_{525}/I_{545}$ under different temperatures were given to fit the calibration curve based on three repeated measurements of the UCL spectra. Error bars were defined as s.d. ($n = 3$). **b** Elevation of A.T. and E.T. of TR-UCNS-labeled cells under irradiation with a 730 nm laser at 140 and 46 mW cm$^{-2}$. Average values of A.T. and E.T. under different time points were given based on three repeated measurements. Error bars were defined as s. d. ($n = 3$). **c** Temperature imaging and luminescence imaging (luminescence of TR-UCNS and DOX) of the TR-UCNS-labeled cells for photothermal-triggered chemotherapy with 730 nm laser irradiation ("730 (−)" and "730 (+)", respectively, indicate the images taken before and after 730 nm laser irradiation. The power density of 730 nm laser used for drug release was 46 mW cm$^{-2}$. Temperature mapping of MIA PaCa-2 cells were acquired according to the thermal equilibrium: $(I_{545})/(I_{525}) = C \exp(-\Delta E/kT)$, where $I_{545}$ and $I_{525}$ were the UCL emission intensities in the wavelength region of 535–580 nm and 515–535 nm, respectively. Scale bars were defined as 30 μm. **d** Temperature imaging and luminescence imaging (luminescence of TR-UCNS and PI) of the TR-UCNS-labeled cells during photothermal therapy with 730 nm laser irradiation ("730 (−)" and "730 (+)", respectively, indicate the images taken before and after 730 nm laser irradiation. The power density of 730 nm laser used for drug release was 140 mW cm$^{-2}$. Scale bars were defined as 30 μm

temperature for drug release in the programmed combination therapy by 730 nm laser at 46 mW cm$^{-2}$), their HSP70 level was identical to the cells treated at 37.0 °C (Supplementary Figure 10a). When the cells were treated at 45.4 °C for 5 min (the same temperature for photothermal therapy in the programmed combination therapy by 730 nm laser at 140 mW cm$^{-2}$), HSP70 level had a very significant increase (Supplementary Figure 10a). Quantitative data showed that HSP70 level of cells treated at 45.4 °C was 5.7-fold higher than cells treated at 41.5 °C (Supplementary Figure 10b). On the other hand, cells treated at 41.5 °C only had a very slight increase of HSP70 expression (~10%), compared to the cells treated at 37.0 °C (Supplementary Figure 10b). These results demonstrated that the generation of HSP affects the therapeutic efficacy of chemodrug, so thus the dosages of chemodrug and heat have to be increased to achieve satisfactory killing effect if they were administrated simultaneously. This also explained that programmed combination therapy has better efficacy under low dosage of drug and heat, since the generation of HSP is negligible with such tiny amount of heat during drug release.

**Programmed combination therapy in vivo.** The excellent treatment effect of the programmed combination therapy by using TR-UCNS prompted us to further carry out the trial of

cancer therapy in vivo. TR-UCNS dispersed in 0.9% NaCl saline (3 mg ml$^{-1}$, 200 μl) were administrated to tumor-bearing mice through intravenous injection. Due to the long circulating property of DPPC, TR-UCNS exhibited a tumor-targeting effect, which was demonstrated by the existence of remarkable UCL signals in the tumor region (Supplementary Figure 11a). Ex vivo bioimaging of the injected mice showed that, except the tumor, TR-UCNS also had a distribution in the liver and the spleen (Supplementary Figure 11b). Histological assays based on hematoxylin and eosin (H&E) staining implied that TR-UCNS had no obvious toxicity in vivo one week after injection (Supplementary Figure 12). In order to determine the laser power densities for drug release and PTT in vivo, UCL signals in the tumor site were collected to decipher the eigen temperature of TR-UCNS. According to the eigen temperature calibration curve measured in a tissue phantom (Fig. 7a), the power densities of 730 nm laser for drug release and PTT in vivo were adjusted to 57 and 164 mW cm$^{-2}$, as the eigen temperature reached to 41.6 and 45.4 °C under such two sets of irradiation conditions (Fig. 7b). Then, mice were divided into five groups (five mice in each group) with different treatments. Mice without any treatment were set as blank group. One group of mice were injected with YSUCNP-PdPc@DPPC (without doxorubicin) and were irradiated by 730 nm laser (57 and 164 mW cm$^{-2}$, 5 min for each power density) to be used for administering PTT only. The other

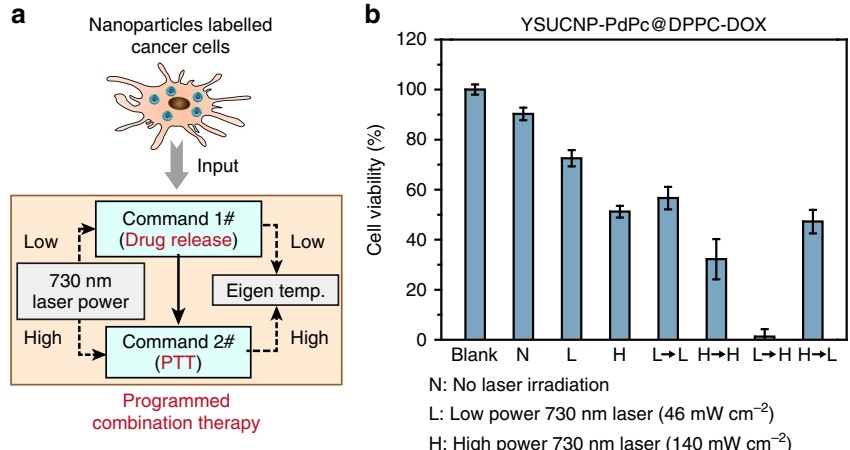

**Fig. 6** Programmed combination therapy at cell level. **a** Scheme of programmed combination therapy. **b** Methyl thiazolyl tetrazolium (MTT) assays of YSUCNP-PdPc@DPPC-DOX-labeled MIA PaCa-2 cells. Cells were treated with no laser irradiation (*N*), 730 nm laser irradiation at 46 mW cm⁻² (*L*), 140 mW cm⁻² (*H*), and their combinations with different sequences (L→L, H→H, L→H and H→L). Each condition in MTT study is tested 3 times and the average value is given. Error bars were defined as s.d. (*n* = 3)

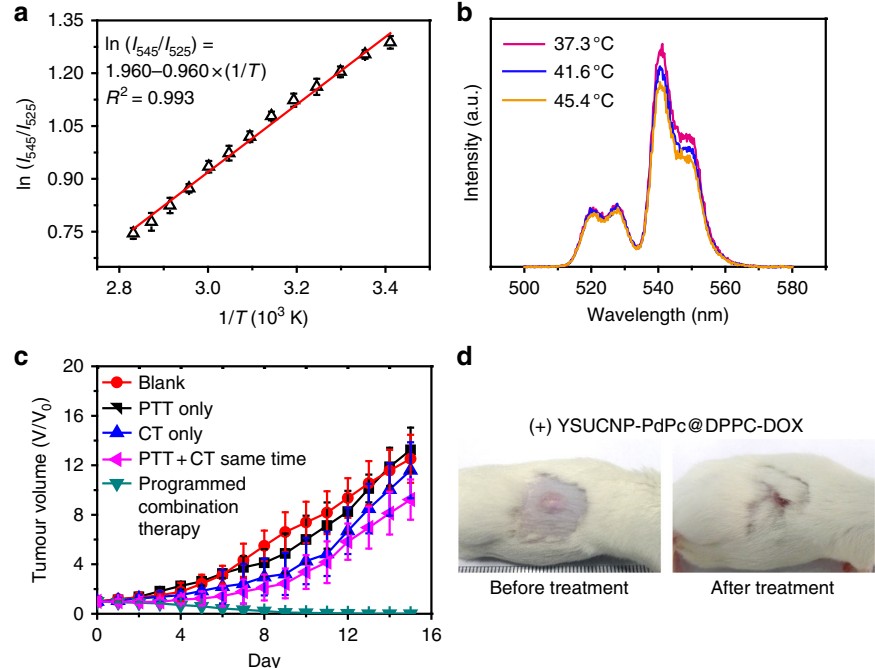

**Fig. 7** Eigen temperature sensing and programfmed combination therapy in vivo. **a** A plot of $\ln(I_{545}/I_{525})$ versus $1/T$ to calibrate the thermometric scale for TR-UCNS in the tissue phantom. Each point was measured 3 times and the average values were used. Error bars were defined as s.d. (*n* = 3).
**b** Upconversion luminescence spectra of TR-UCNS (500–580 nm) in tumor-bearing mice detected by fiber-optic spectrometer under different 730 nm laser power density at 0, 57, and 164 mW cm⁻². Eigen temperature calculated by the upconversion luminescence spectra were 37.3, 41.6, and 45.4 °C.
**c** Growth of tumors in different groups of mice (*n* = 5 in each group) after treatment. The relative tumor volumes were normalized to their initial sizes. Average values of the tumor volume of each group were based on the measurement of the tumors in five mice. Error bars were defined as s.d. (*n* = 5).
**d** Representative photos of tumor-bearing mice with programmed combination therapy

group of mice injected with TR-UCNS and irradiated by 730 nm laser (57 mW cm⁻², 10 min) were used for administering CT only. Another group was injected with TR-UCNS and irradiated by 730 nm laser (164 mW cm⁻², 10 min) for administering PTT and CT simultaneously, imitating the conventional combination therapy. Finally, another group of mice were irradiated by 730 nm laser with two sets of power density (57 mW cm⁻² and then 164 mW cm⁻², 5 min for each power density) to perform the programmed combination therapy. Laser irradiation was performed once a day for a total of 5 days. The tumors in the groups of Blank, PTT only, and CT only grew continuously (Fig. 7c) because the dosage of DOX and heat were rather low. PTT and CT performed simultaneously (conventional combination therapy) showed a limited therapeutic effect in five days and the tumors started to grow gradually in the following days. This is probably because a portion of the cancer cells were killed at the beginning, while the remaining portion of cells survived, and thus the tumors growth was not suppressed later. In the group treated

with programmed combination therapy, however, the tumors shrunk within 6 days and completely eliminated (Fig. 7c,d, Supplementary Figure 13 and Supplementary Figure 14), and all five mice survived in forty days of the viability test (Supplementary Figure 14). These results indicate that programmed combination therapy can maximize the utility of the two therapeutic modes with low dosages of heat and chemodrug, and consequently, achieve the ideal therapeutic effect where the reoccurrence of tumors is avoided.

## Discussion

In this work, we presented a temperature responsive upconversion nanosystem, NaLuF$_4$:20%Yb,2%Er@NaLuF$_4$@YS-SiO$_2$-PdPc@DPPC-DOX (TR-UCNS), which contains both chemodrug and photothermal agent. Under the excitation of near-infrared 730 nm laser, TR-UCNS can generate photothermal effect and achieve thermal triggered drug release for photothermal therapy and chemotherapy. Due to thermal sensitive nature of the upconversion luminescence generated from TR-UCNS, microscopic temperature of the nanocomposite during photothermal process is determined. By controlling the microscopic temperature, the sequences of chemotherapy and photothermal therapy can be deliberately arranged to achieve programmed combination cancer therapy. When the dosages of chemodrug and heat are kept at low level (2.5 μM of DOX and heat generated by ~150 mW cm$^{-2}$ of 730 nm laser), programmed combination therapy can achieve 39 folds improvement (Fig. 6b) in therapeutic effect in vitro than conventional combination therapy that initiates chemotherapy and photothermal therapy at the same time. This indicates that programmed combination therapy uses less drug and heat to realize an ideal killing effect that conventional combination therapy may reach by using more drug and heat. It is worth noting that if the same therapeutic effect as programmed combination therapy is wanted, up to 8 folds of DOX is needed (20 μM) or up to 2.6 folds of 730 nm laser power density is needed (400 mW cm$^{-2}$) (Supplementary Figure 7a and b). Furthermore, the therapeutic effect of programmed combination therapy was also proved in tumor bearing mice. This work provides a non-invasive method to precisely modulate the process of combination therapy which circumvents the interference between the treatment modes and thus maximizes the utility of each treatment mode. Consequently, the applied dosage can be lowered and the therapeutic efficacy is improved at the same time. With the ability to apply such programming methods into other combination therapy strategies, it will allow the medical fraternity to create personalized medical options. The results of this work extend the application of the microscopic temperature modulation and open up a new opportunity for designing next generation of cancer therapy strategies.

## Methods

**Synthesis of NaLuF$_4$:Yb,Er@NaLuF$_4$ (csUCNPs).** Typically, 1 mmol LnCl$_3$ (Ln: Lu, Yb, Er) with the molar ratio of 78:20:2 (for the synthesis of NaLuF$_4$:Yb,Er nanoparticles) were added to a 100 mL three-necked flask containing 6 mL OA and 15 mL ODE. The mixture was heated to 90 °C for degassing and then heated to 150 °C for 30 min to obtain a transparent solution. The solution was cooled to 50 °C afterwards and NH$_4$F (4 mmol) and NaOH (2.5 mmol) dissolved in 5 ml methanol was added into the solution. After degassing for 20 min at 90 °C, the mixture was heated to 300 °C as soon as possible and kept at this temperature under argon for 1 h. When the mixture was cooled down to room temperature, the nanoparticles were precipitated by pouring 20 mL ethanol and cyclohexane (1:1) solution, and collected by centrifugation (15,900×g, 8 min). After washing with ethanol and cyclohexane for three times, hexagonal phase NaLuF$_4$:20%Yb,2%Er nanoparticles were finally redispersed in cyclohexane. NaLuF$_4$:Yb,Er@NaLuF$_4$ nanoparticles were prepared by epitaxial growth of NaLuF$_4$ layer on NaLuF$_4$:Yb,Er through a similar procedure. A total of 1.0 mmol LuCl$_3$ was added into a mixed solution of 8 mL oleic acid (OA) and 12 mL octadecene (ODE). The mixture was degassed at 90 °C and then heated to 150 °C for 30 min to obtain a transparent solution. After that, 3 mL ODE containing the as-prepared NaLuF$_4$:Yb,Er nanocrystals was mixed with

the solution. The temperature of the mixture was gently raised to 60 °C to remove cyclohexane. The following procedures were the same as the synthesis of NaYF$_4$:Yb,Er nanoparticles. The as-obtained NaLuF$_4$:Yb,Er@NaLuF$_4$ nanoparticles were dispersed in 10 ml cyclohexane for further use.

**Synthesis of NaLuF$_4$:Yb,Er@NaLuF$_4$@YS-SiO$_2$ (YSUCNP).** Igepal CO-520 (0.5 g) was dissolved in 5 ml cyclohexane to form reaction solution A. csUCNPs (10 mg) were dispersed in 5 ml cyclohexane and ultrasonicated for 5 min to form reaction solution B. Then, A and B were mixed together with vigorous stirring. After stirring for 1 h, 50 μl of ammonia solution (25%) was added dropwise and the mixture was stirred continuously for another 3 h. After that, 15 μl of TEOS was added into the solution on the first day, 30 μl of TEOS and 5 μl TSD were added on the second day, and then 30 μl of TEOS was added on the third day to achieve a multi-layered silica coating. The 10 ml of ethanol was poured into the solution to precipitate the resulting nanoparticles. The products were collected by centrifugation and washed by ethanol 5 times. To make the nanoparticles with a yolk-shell-like structure, the as-obtained nanoparticles were dispersed into water/ethanol (1:1 v/v%) mixed solution and then 1 ml of 1% HF solution (wt%) was added and stirred for 6 h at room temperature. Then, YSUCNP were collected and washed by ethanol 3 times and finally dispersed in ethanol for further use.

**Synthesis of YSUCNP-PdPc.** To synthesize octabutoxyphthalocyanine palladium (II) (PdPc), 0.1 mmol 1, 4, 8, 11, 15, 18, 22, 25-octabutoxy-29H,31H-phthalocyanine and 0.3 mmol PdCl$_2$ were added into 5 ml anhydrous dimethylformamide. The mixture was heated to 130 °C with the protection of N$_2$ for 24 h. When the mixture was cooled down to room temperature, 5 ml methanol was added to precipitate PdPc. The product was washed by 10 ml methanol to remove excessive PdCl$_2$ and then dried for further use. The 20 mg YSUCNP and 2 mg PdPc were added to 10 ml chloroform. The mixture was ultrasonicated at 30 °C for 5 min and then the solvent was removed by a rotary evaporator. After that, 10 ml water was added and YSUCNP-PdPc were redispersed by ultrasonication. The obtained YSUCNP-PdPc was washed with ethanol for one time and water for two times. The adsorbing quantity of the YSUCNP was determined by UV-Vis absorption spectrum. The standard solution of PdPc was prepared and a calibration curve was obtained, and then the supernatant of YSUCNP-PdPc was measured. The photothermal agent (PdPc) loaded on 20 mg YSUCNP is 1.53 mg (2 mg × 76.4%, Supplementary Figure 3d).

**Synthesis of YSUCNP-PdPc@DPPC-DOX (TR-UCNS).** Firstly, DOX-HCl was neutralized by NaOH aqueous solution to precipitate hydrophobic DOX. The resulting hydrophobic DOX was dissolved in ethanol to form a solution with a concentration of 2.5 mM. Then, 10 mg of DPPC and 50 μl of DOX solution were dissolved in 10 ml chloroform. The solvent was removed by a rotary evaporator and formed a homogeneous membrane at the bottom of the bottle. Then, 10 ml aqueous solution of YSUCNP-PdPc (1 mg ml$^{-1}$) was added and the mixture was ultrasonicated to form YSUCNP-PdPc@DPPC-DOX. The obtained YSUCNP-PdPc@DPPC-DOX was separated by centrifugation (7200×g, 8 min) and washed with water for three times. According to the absorption data of DOX in Supplementary Figure 3e, 0.115 μmol (2.5 mM × 50 μl × 91.8%) of DOX molecules are loaded in 10 mg of YSUCNP-PdPc (10 mg YSUCNP-PdPc contain 9.3 mg YSUCNP.). For the convenience of quantification and simple expression in the following experiments, the dosages of TR-UCNS in the solution for temperature detecting, cell incubation, and in vivo administration are represented by the mass concentration of YSUCNP contained in TR-UCNS.

**Measurement of eigen temperature of TR-UCNS in the solution.** The calibration curve of temperature and upconversion luminescence was obtained in our designed system[56]. TR-UCNS aqueous dispersion (2 ml, 0.5 mg ml$^{-1}$, mass concentration here is represented by YSUCNP in TR-UCNS) in a quartz cuvette was put in Edinburgh FLS-920 fluorescence spectrometer with a temperature adjustable holder. The solution was controlled at different temperatures from 10 to 90 °C, and the upconversion luminescence from 500 to 580 nm at each temperature point was collected under the excitation of a continuous wave (CW) 980 nm laser (50 mW cm$^{-2}$). The intensity ratio of the UCL centered at 525 and 545 nm at each temperature point was substituted in Eq. 1 to get the calibration curve. Then, the solution was irradiated by 730 nm laser at different power density. UCL spectra of TR-UCNS were collected under 980 nm laser excitation to calculate the eigen temperature, and the apparent temperature was detected by a thermocouple.

**Observation of DOX release in TR-UCNS.** The release behavior of DOX in TR-UCNS was observed by recording the fluorescence of DOX under 488 nm light excitation. Aqueous dispersion containing TR-UCNS (2 ml, 0.5 mg ml$^{-1}$, mass concentration here is represented by YSUCNP in TR-UCNS) was placed in Edinburgh FLS-920 fluorescence spectrometer. The solution was heated to different temperatures from 32 to 48 °C to figure out the temperature threshold (41.6 °C) for the drug release. On the other hand, another solution was irradiated by 730 nm laser to investigate the release behavior under photothermal effect. Eigen temperature of TR-UCNS was set at ~41.6 °C by controlling the laser power density at

$50\ mW\ cm^{-2}$. The corresponding apparent temperature was 37.8 °C. The release behavior at 37.8 and 37 °C was also observed. Fluorescence spectra of DOX excited by 488 nm light of xenon lamp were collected at each time point to calculate the release rate.

**Cell culture and confocal upconversion imaging.** MIA PaCa-2 cells were provided by the Institute of Biochemistry and Cell Biology, SIBS, CAS (China). MIA PaCa-2 cell line used in this work is not included in the list of misidentified cell lines made by International Cell Line Authentication Committee (ICLAC) and is tested without mycoplasma contamination. Cells ($1 \times 10^8\ l^{-1}$) were plated on culture dish incubated with DMEM (Dulbecco's Modified Eagle Medium) supplemented by 10% fetal bovine serum at 37 °C and 5% $CO_2$ for 24 h. Then, the cells were washed with phosphate buffer and 2 ml of a serum-free medium containing $200\ \mu g\ ml^{-1}$ (mass concentration is represented by YSUCNP in TR-UCNS. Dosage of DOX and PdPc contained are 2.5 μM and $15.3\ \mu g\ ml^{-1}$, respectively.) TR-UCNS was added into the dish for further incubation at 37 °C and 5% $CO_2$ for 2 h. After that, the cells were washed with phosphate buffer three times to perform confocal microscopy imaging. Confocal upconversion imaging was accomplished on the laser scanning UCL microscope with an Olympus FV1000 scanning unit, which is designed by our group[69]. UCL signals were excited by a CW laser at 980 nm with a focused power of 19 mW. The UCL signals at 500–580 nm were collected under a 60× oil-immersion objective lens. To observe the release of DOX and therapeutic efficacy, cells were irradiated by a 730 nm laser at $46\ mW\ cm^{-2}$ first and at $140\ mW\ cm^{-2}$. The irradiation time under each power density was 5 min. Propidium iodide (PI) was used to identify whether the cells were dead after each time of irradiation. Fluorescence signals of DOX were collected at 550–610 nm under 488 nm excitation. PI signals were collected at 650–700 nm under 532 nm excitation.

**Evaluation of cell viability.** Cell viability was quantitatively evaluated by methyl thiazolyl tetrazolium (MTT) assays. Briefly, the cells were planted into a 96-well cell culture plate at $5 \times 10^4$ per well and were cultured at 37 °C and 5% $CO_2$ for 24 h. After that, the cells were treated with different conditions (including the incubation of DOX, TR-UCNS at different concentration, and laser irradiations under different power densities) and further cultured at 37 °C and 5% $CO_2$ for another 24 h. Then, MTT ($5\ \mu l$, $5\ mg\ ml^{-1}$) was added to each well and the cells were incubated at 37 °C and 5% $CO_2$ for additional 4 h. A total of 50 μl of 10% SDS was added in each well and the plate was held at room temperature for 12 h. The optical density $OD_{570}$ value (Abs.) of each well, with background subtraction at 690 nm, was detected by a Tecan Infinite M200 monochromator-based multifunction microplate reader. Cell viability (%) was calculated using Eq. 2, which is given below:

$$Cell\ viability(\%) = \left(\frac{Mean\ of\ Abs.\ value\ of\ treatment\ group}{Mean\ Abs.\ value\ of\ control}\right)100\% \qquad (2)$$

**HSP70 expression analysis.** Western blot analysis was used to determine the expression of HSP70 in MIA PaCa-2 cells. Two groups of cells were treated at 41.5 or 45.4 °C for 5 min, which simulate the drug release and photothermal process in programmed combination therapy. The other set of cells was treated at 37.0 °C as control. For western blot experiment, an equal amount of total protein was added in each well for electrophoresis. The separated protein was transferred into polyvinylidene difluoride membrane from the gel. β-Actin in the cells was used as internal control and HSP70 was used to represent the heat shock protein. Anti-HSP70 antibody [C92F3A-5] (ab47455, Abcam) was used as primary antibody for HSP70 determination. Anti-β-Actin antibody (ab8226, Abcam) was used as primary antibody for β-Actin determination. The ratio of primary antibodies for β-Actin and HSP70 is 1:1000 and the ratio of secondary antibodies is 1:10,000. The protein bands were scanned digitally for data representation and quantitative analysis. For quantitative analysis, each group of samples (37.0, 41.5, and 45.4 °C) was in triplicate.

**Eigen temperature mapping of TR-UCNS-labeled cells.** Temperature calibration curve for cell was measured in laser scanning UCL microscope with an Olympus FV1000 scanning unit and the temperature was adjusted with help of an external heating platform. UCL signals at 540–570 nm ($I_{545}$) and 515–535 nm ($I_{525}$) were collected, respectively, at certain temperature to determine the ratio. Cells incubated with TR-UCNS were irradiated by 730 nm laser at $46\ mW\ cm^{-2}$ or $140\ mW\ cm^{-2}$ for 5 min. The corresponding UCL images at 540–570 nm and 515–535 nm during 730 nm laser irradiation were collected and manipulated mathematically based on the calibration formula to obtain the image of the eigen temperature.

**Tumor xenografts.** Animal procedures were in accordance with the guidelines of the Institutional Animal Care and Use Committee (IACUC), School of Pharmacy, Fudan University. MIA PaCa-2 cells were harvested by incubation with 0.05% trypsin-EDTA and then collected by centrifugation and resuspended in sterile phosphate buffer saline. Cells ($10^8$ cells per mouse) were subcutaneously implanted into four-week-old male Balb/c scid mice. Tumor-bearing mice were ready for bioimaging, and programmed combination therapy was performed when the tumors reached an average diameter of 0.5 cm.

**Upconversion bioimaging in vivo.** Whole-body bioimaging was accomplished by the in vivo imaging system designed by our group[70]. A 0–5 W adjustable continuous wave 980 nm laser was used as excitation and Andor DU897 EMCCD was used as a signal collector. A 720 nm short-pass filter was used to exclude the excitation light. Tumor-bearing mice were injected with 0.9% NaCl saline containing TR-UCNS ($3\ mg\ ml^{-1}$, 200 μl) through the tail vein. UCL was performed 6 h after injection of TR-UCNS.

**Detection of eigen temperature in vivo.** The temperature calibration curve for living body was obtained in tissue phantom of which the synthetic method is reported elsewhere. A 2 $cm^3$ tissue phantom containing TR-UCNS ($0.5\ mg\ cm^{-3}$, represented by the mass concentration of YSUCNP contained in TR-UCNS) was put in quartz cuvette and heated to a series of temperature from 10–90 °C. UCL spectra at each temperature point were recorded by Edinburgh FLS-920 fluorescence spectrometer. To observe the eigen temperature during photothermal process, tissue phantom was irradiated by a 730 nm laser, and the corresponding UCL spectra were collected to calculate the eigen temperature, according to the calibration curve. As to the eigen temperature monitoring in tumor-bearing mice, a fiber-optic spectrometer (PG2000 Pro, Ideaoptics, China) was used to collect the UCL signals at the tumor site under excitation of 980 nm laser. The 730 nm laser power densities for drug release and photothermal therapy were determined according to the eigen temperature measured in vivo.

**Programmed combination therapy in vivo.** For programmed combination therapy, mice injected with TR-UCNS were irradiated by the 730 nm laser with a light spot of ~10 mm diameter and was focused on the tumor area. Mice were first irradiated at the power density of $57\ mW\ cm^{-2}$ for 5 min and then irradiated at $164\ mW\ cm^{-2}$ for another 5 min after 4 h from the first time of irradiation. The control groups were irradiated at $57\ mW\ cm^{-2}$ for 10 min or at $164\ mW\ cm^{-2}$ for 10 min. The blank group was not irradiated at 730 nm laser. Tumor sizes of each group were measured every day after treatment. Each group contained five mice. The tumor sizes were measured using a caliper and calculated as volume = (tumor length) × (tumor width)$^2$/2. Relative tumor volumes were normalized and were calculated as $V/V_0$ ($V_0$ is the tumor volume when the treatment was initiated) and average values were used. Error bars were defined as s.d. No blinding or randomization was used in animal studies. According to the guidelines of Institutional Animal Care and Use Committee (IACUC), School of Pharmacy, Fudan University, the maximum permitted tumor size is 20 mm in an average diameter for mice. The tumors' size in this work are confined within this criterion.

**Other information.** The source of all law materials, details of characterization, and calculation of the photothermal conversion efficiency are presented in the Supplementary Methods section in Supplementary Information file.

**Data availability.** The authors declare that all the data supporting the findings of this study are presented in the article and its Supplementary Information files. All relevant data are also available from the corresponding author upon request.

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

## Acknowledgements

We thank the National Natural Science Foundation of China (21527801, 21231004, and 21722101), the National Key R&D Program of China (Grant 2017YFA0205100), the National Basic Research Program of China (2015CB931800), and Shanghai Sci. Tech. Comm. (15QA1400700) for financial support. We also thank Dr. Jianfeng Li for the discussion and valuable suggestions on this work.

## Author contributions

The manuscript was written through the contributions of all the authors. All authors have given approval to the final version of the manuscript. X.Z., J.L., X.Q., and Y.L. performed the experiments. X.Z., W.F., and F.L. analyzed the data, wrote the manuscript, and designed the experimental approach.

## Additional information

**Competing interests:** The authors declare no competing interests.

