## [Peer Review File · Nature Communications]

Reviewers' Comments:

Reviewer #1:

Remarks to the Author:

In this manuscript, the authors, using upconversion luminescent technology to monitor the microscopic temperature (eigen temperature) of nanocomposite, come up with the novel concept of programming combination cancer therapy by precise control of microscopic temperature, in which lower laser power density is used for chemotherapy drug release while a higher one is used for photothermal therapy. The rational experiment design, solid data, and significant conclusion will contribute to the rapid development of cancer treatment in the future, which is definitely of interest and value to the chemistry and nanomaterial communities. In my opinion, this manuscript should be considered for acceptance after minor modifications. My comments are as follows:

1. In the end of part 1 of Results, the authors should summarize the composition of the YSUCNP-PdAPc@DPPC-DOX, especially, the DOX content and PdPc in each particles, which can be measured by TGA analysis of YSUCNP-PdPc and YSUCNP-PdPc@DPPC-DOX, combined with their results in Supplementary Figure 3e.

2. For Figure 6a, does the interval between Command 1 and Command 2 have any effect on the treatment efficacy?

3. In Line 293, the authors should give the related reference.

4. In page 19, concerning the mechanic studies, the authors should analyze the level of HSP in the different laser (46mW and 140 mW) processed cell samples by Western Blotting method.

5. A challenging question for authors: Does singlet oxygen contribute to the treatment from the photodynamic sensitizer and laser irradiation used in the current treatment?

6. In Supplementary Figure 9, the authors should label pre-treatment and post-treatment, and label the organs.

7. Errors in the manuscript:

Line 207 and 209, 46 W and 140 W should be 46 mW and 140 mW;

Line 226, DOX should be PI.

Reviewer #2:

Remarks to the Author:

The present work aims at optimizing combined cancer therapy (chemotherapy, CT, and photothermal therapy, PTT) by designing a clever nanosystem including a nanothermometer (upconverting lanthanide-based nanoparticle, UCNP), a chemical drug (doxorubicin), and a photothermal agent (palladium(II) phthalocyanine).

The main finding of the study lies in demonstrating that sequential application of the two treatments (CT and PTT) has a far better efficiency than simultaneous application as it is usually practiced. The trick here is to monitor the temperature of the cells via the UCNP, excited at 980 nm), which subsequently allows one to tune the light fluence and irradiation dose at 730 nm to generate CT, and then PTT. The advantage of the sequential treatment is first demonstrated in vitro on MIA PaCa-2 cells and finally in vivo on Balb/c mice. Additionally, another important finding is that generation of heat shock protein (HSP) is much reduced in the sequential treatment as opposed to the simultaneous protocol, meaning that lower dosage of chemodrug and photothermal agent can be used, reducing side effects.

All data are reported with standard deviations and satisfying statistics is provided for the in vivo experiments (5 mice for each of the cohorts). Other experimental procedures are described with necessary details, in particular the morphology of the nanocomposites.

This contribution represents an important advance in the field of cancer therapy and will have a broad impact in both the biomedical and photophysics communities. Conclusions are convincing and at this stage, further experiments are not required, as the principle is well established. It will of course have to be checked for each specific case (cancer type, nature of chemodrug and photothermal agent) but this is clearly out of the scope of such a communication.

As a conclusion, I recommend publication after some polishing of the manuscript (and, also some

clarifications).

- For instance the introduction is somewhat long with repetitions, the role of the two wavelengths used (980 for T measurements and 730 nm for therapy, with low – CT – and high – PTT – power densities) should be described in a clearer way.
- Caption to Fig. 3. Please give Er(III) concentration and, also, excitation wavelength
- There is probably an error in reporting power densities used for the in vitro study (46 and 140 W/cm²) in the text (lines 207, 209), while corresponding captions to figures mention mW.
- There is some confusion in the description of the in vitro experiments. While at the beginning (lines 229-230) it is clearly stated that all cell experiments were conducted with MIA PaCa-2 cells, Figs. 5c/d report images of HeLa cells (as said in the caption). If different cell lines were used, please explain why and, also, clearly state this in the captions to Fig. 5.
- Caption to Figure 5. Please explain the meaning of 730 nm (-) and 730 nm (+)
- Figure 7c. Please give the dosages of drug and photothermal agent used
- Line 352. Can the authors quantify what they mean by "less dosages"? Which proportion? Getting this information would strengthen the article.

Some suggested text improvements

- Line 35, replace "received" with "resulted in"
- Line 39, replace "nanocomposite" with "a nanocomposite"
- Line 42, replace "have" with "has"
- Line 85, replace "a n on-doping" with "an undoped"

J.-C. Bünzli

Responses to the Reviewers' Reports Manuscript (NCOMMS-17-419223)

Reviewer: 1

Comments:

In this manuscript, the authors, using upconversion luminescent technology to monitor the microscopic temperature (eigen temperature) of nanocomposite, come up with the novel concept of programming combination cancer therapy by precise control of microscopic temperature, in which lower laser power density is used for chemotherapy drug release while a higher one is used for photothermal therapy. The rational experiment design, solid data, and significant conclusion will contribute to the rapid development of cancer treatment in the future, which is definitely of interest and value to the chemistry and nanomaterial communities. In my opinion, this manuscript should be considered for acceptance after minor modifications. My comments are as follows:

Response: We greatly appreciate the carefully reading and the positive comment from the reviewer.

Comment 1. In the end of part 1 of Results, the authors should summarize the composition of the YSUCNP-PdAPc@DPPC-DOX, especially, the DOX content and PdPc in each particles, which can be measured by TGA analysis of YSUCNP-PdPc and YSUCNP-PdPc@DPPC-DOX, combined with their results in Supplementary Figure 3e.

Response: We greatly appreciate this valuable advice. TGA analysis is indeed a commonly used method to determine the composition of nanomaterials. However, TGA analysis is not suitable to be used here for characterizing the nanocomposite in this work since TR-UCNS has multiple components (PdPc, DPPC and DOX) and the percentage of each component is hard to be told by weight loss. We are thankful to the reviewer for helping us improve our work by raising this suggestion. As an alternative solution, the approximate number of PdPc and DOX in each nanoparticle can be clearly given by computing the data in Supplementary Figure 3d and 3e.

The densities of NaLuF₄:20% Yb,2% Er, NaLuF₄ and SiO₂ are 6.49, 6.52 and 2.40 g cm⁻³, respectively. The diameters of NaLuF₄:20% Yb,2% Er and NaLuF₄:20% Yb,2% Er@NaLuF₄ are 13.4 and 20.5 nm. The external and internal diameters of yolk-shell silica layer are 48.1 and 36.5 nm. Hence, the total weight of one

NaLuF₄:20% Yb,2% Er@NaLuF₄@Yolk-Shell SiO₂ (YSUCNP) is 8.10×10^{-17} g, that is $\{[(24.05 \text{ nm})^3 - (18.25 \text{ nm})^3] \times 2.40 \text{ g cm}^{-3} + [(10.25 \text{ nm})^3 - (6.70 \text{ nm})^3] \times 6.52 \text{ g cm}^{-3} + (6.7 \text{ nm})^3 \times 6.49 \text{ g cm}^{-3}\} \times \pi \times 10^{-17}$. The molar of 20 mg YSUCNP is 0.41 nmol. According to the data in Supplementary Figure 3d, the molar of PdPc combined with 20 mg YSUCNP is 1.28 μmol ($2 \text{ mg} \times 76.4\% \times 1197.8 \text{ g mol}^{-1}$). Hence, there are ~3120 (calculated by $1.28 \mu\text{mol} / 0.41 \text{ nmol}$) PdPc molecules in each YSUCNP particle. Based on the data in Supplementary Figure 3e, DOX laden on 10 mg YSUCNP@PdPc (which contain 0.19 nmol YSUCNP nanoparticles) is 0.115 μmol ($2.5 \text{ mM} \times 50 \mu\text{l} \times 91.8\%$), so there are ~605 (calculated by $0.115 \mu\text{mol} / 0.19 \text{ nmol}$) DOX molecules on each nanoparticle. The description of the approximate number of PdPc and DOX on each nanoparticles is added in the second paragraph of “**Result: Synthesis and characterizations of TR-UCNS.**” of the manuscript, which is labeled in yellow.

Please see the added contents below,

“According to the data shown in Figure S3d and S3e, the molar of PdPc loaded in 20 mg YSUCNP is 1.28 μmol ($2 \text{ mg} \times 76.4\% \times 1197.8 \text{ g mol}^{-1}$). The molar of 20 mg YSUCNP is 0.41 nmol. The number of PdPc in each YSUCNP nanoparticle is calculated by $1.28 \mu\text{mol} / 0.41 \text{ nmol}$. Hence, there are ~3120 PdPc molecules in each YSUCNP particle. DOX laden on 10 mg YSUCNP@PdPc (which contain 0.19 nmol YSUCNP nanoparticles) is 0.115 μmol ($2.5 \text{ mM} \times 50 \mu\text{l} \times 91.8\%$), so there are ~605 (calculated by $0.115 \mu\text{mol} / 0.19 \text{ nmol}$) DOX molecules on each nanoparticle.”

Comment 2. For Figure 6a, does the interval between Command 1 and Command 2 have any effect on the treatment efficacy?

Response: We appreciate the reviewer for this question. The interval between Command 1 and 2 were optimized in the programmed combination therapy presented in this work. Four hours' interval between Command 1 and 2 reached a maximized efficacy for programmed combination therapy. The data for choosing the interval time was added in the Supplementary information of the revised manuscript as “Supplementary Figure 8”, which is shown below. The description of the time interval between Command 1 and 2 was supplemented in the manuscript, “**Results: Programmed combination therapy in vitro.**”, Paragraph 2 and labeled in yellow.

Supplementary Figure 8. Viability of MIA PaCa-2 cells treated with programmed combination therapy with different time intervals between Command 1 (Drug release) and Command 2 (Photothermal therapy). The power density of 730 nm laser for drug release and photothermal therapy are 46 mW cm^{-2} and 140 mW cm^{-2} , respectively. Four hours' interval between Command 1 and 2 resulted in the maximized therapeutic effect so that this time interval was employed for in vitro and in vivo programmed combination therapy study.

Comment 3. In Line 293, the authors should give the related reference.

Response: We appreciate this valuable advice. The references (Ref. 64-68) have been supplemented to support the statement of heat shock protein induced by thermotherapy will protect the cancer cells from the killing effect of chemodrug.

Please see the supplemented references below,

64 Pelz, J. O. W. *et al.* Hyperthermic Intraperitoneal Chemotherapy in Patients with Peritoneal Carcinomatosis: Role of Heat Shock Proteins and Dissecting Effects of Hyperthermia. *Annals of Surgical Oncology* **20**, 1105-1113, (2013).

65 Ciocca, D. R. *et al.* Response of human breast cancer cells to heat shock and chemotherapeutic drugs. *Cancer research* **52**, 3648-3654, (1992).

66 Engelhardt, R. in *Hyperthermia and the therapy of malignant tumors* 136-203 (Springer, 1987).

67 Donaldson, S., Gordon, L. & Hahn, G. Protective effect of hyperthermia against the cytotoxicity of actinomycin D on Chinese hamster cells. *Cancer treatment reports* **62**, 1489-1495, (1978).

68 Wallner, K. & Li, G. C. Adriamycin resistance, heat resistance and radiation

response in Chinese hamster fibroblasts. *International Journal of Radiation Oncology* Biology* Physics* **12**, 829-833, (1986).”

Comment 4. *In page 19, concerning the mechanic studies, the authors should analyze the level of HSP in the different laser (46mW and 140 mW) processed cell samples by Western Blotting method.*

Response: We appreciate this valuable advice. To give more evidence for the possible mechanism of programmed combination therapy, we used western blot analysis to evaluate the expression level of heat shock protein 70 (HSP70) of the cells with different treatment. MIA PaCa-2 cells were treated at 41.5 and 45.4 °C for DOX release and photothermal therapy in programmed combination therapy, respectively. As shown in Supplementary Figure S10, when cells were treated at 41.5 °C for 5 min (the same temperature for drug release in programmed combination therapy by 730 nm laser at 46 mW cm⁻²), their HSP70 level was identical to the cells treated with 37.0 °C. However, when the cells were treated with 45.4 °C for 5 min (the same temperature for photothermal therapy in programmed combination therapy by 730 nm laser at 140 mW cm⁻²), the HSP70 expression increased significantly. Quantitative data showed that HSP70 level of cells treated with 45.4 °C was 5.7 folds higher than cells treated with 41.5 °C. On the other hand, cells treated with 41.5 °C only had a very slight increase of HSP70 expression (~10%) compared to the cells treated with 37.0 °C. These results implied that high initial temperature as used in conventional combination therapy method will cause significant increase of heat shock protein, while the generation of HSP70 will reduce the therapeutic effect of combination therapy which is shown in Supplementary Figure 9. As a result, we can further confirm that the effectiveness of programmed combination therapy in a low drug and heat dosages is very likely attributed to the circumvention of heat shock protein generation.

Western blot analysis data is presented as Supplementary Figure 10, which is shown below,

Supplementary Figure 10. (a) HSP70 expressions of MIA CaPa-2 cells under 37.0, 41.5 and 45.4 °C treatment with western blot analysis. β -Actin was used as internal control. (b) Quantitative data of the expression of HSP70 in MIA CaPa-2 cells under 37.0, 41.5 and 45.4 °C treatment.

Comment 5. A challenging question for authors: Does singlet oxygen contribute to the treatment from the photodynamic sensitizer and laser irradiation used in the current treatment?

Response: Thanks for this question. We used a reactive oxygen species probe (ROS), furfuryl alcohol (FFA) (*Environ. Sci. Technol.* 20, 341-348 (1986); *Environ. Sci. Technol.* 44, 6674-6679 (2010)), to detect the amount of ROS generated by YSUCNP-PdPc@DPPC during 730 nm laser irradiation. Aqueous solution containing 0.4 mg ml⁻¹ YSUCNP-PdPc@DPPC and 0.006 mol l⁻¹ FFA aqueous solution was irradiated by 730 nm laser at 150 mW cm⁻² for 20 min. As shown in Figure R1b, compared to the control group without 730 nm laser irradiation, there is only 4.1% decrease in the absorption peak of FFA, which indicates a nearly neglectable ROS generation. It should be noted that the power density of 730 nm laser (150 mW cm⁻²) for photodynamic effect evaluation is the highest power density used for programmed combination therapy in our work. Hence, the photodynamic effect in our work is not necessary to be taken into consideration. The laser power density for photodynamic therapy with phthalocyanine is about 1 W/cm² as reported by our previous work (*Biomaterials* 34, 7905-7912, (2013)).

Figure R1. (a) Relative curve of the concentration of furfuryl alcohol (FFA) versus the absorption value at 273 nm. (b) Absorption of spectra of FFA aqueous solution with (Blue line) and without (Pink line) 730 nm irradiation. FFA solution was mixed with YSUCNP-PdPc@DPPC first and then irradiated with 730 nm laser at 0 or 150 mW cm⁻² for 20 min. FFA solution was centrifuged to remove YSUCNP-PdPc@DPPC before absorption test.

Comment 6. In Supplementary Figure 9, the authors should label pre-treatment and post-treatment, and label the organs.

Response: Thanks very much for this helpful advice. We added labels in Supplementary Figure 11 (Originally Supplementary Figure 9) to distinguish the bright field from overlay images and also annotated the organs for easy recognition. Please see the revised version below,

Supplementary Figure 11. Tumor targeted upconversion luminescence imaging. **(a)** *In vivo* UCL imaging of tumor-bearing Balb/c mice 6 h after intravenous injection of TR-UCNS (2 mg ml^{-1} , $200 \mu\text{L}$). Left, bright field image of tumor-bearing Balb/c mice. Right, overlay of bright field and UCL images. **(b)** *Ex vivo* UCL image of the major organs of the injected mice (heart, lung, spleen, liver, kidney and tumor). Left, bright field image of tumor-bearing Balb/c mice. Right, overlay of bright field and UCL images. UCL signals were collected by using a 720 nm short pass filter.

Comment 7. Errors in the manuscript:

Line 207 and 209, 46 W and 140 W should be 46 mW and 140 mW;

Line 226, DOX should be PI.

Response: We appreciate this valuable advice and deeply apologize for the errors in the manuscript. The unit of 730 nm laser power density described in “**Results: Programmed combination therapy in vitro.**” (Originally Line 207 and 209) were corrected into “ mW cm^{-2} ”. The mistaken word of “DOX” in the caption of Fig. 5d (Originally Line 226) was also corrected into “PI”.

Reviewer: 2

Comments:

The present work aims at optimizing combined cancer therapy (chemotherapy, CT, and photothermal therapy, PTT) by designing a clever nanosystem including a nanothermometer (upconverting lanthanide-based nanoparticle, UCNP), a chemical drug (doxorubicin), and a photothermal agent (palladium(II) phthalocyanine).

The main finding of the study lies in demonstrating that sequential application of the two treatments (CT and PTT) has a far better efficiency than simultaneous application as it is usually practiced. The trick here is to monitor the temperature of the cells via the UCNP, excited at 980 nm), which subsequently allows one to tune the light fluence and irradiation dose at 730 nm to generate CT, and then PTT. The advantage of the sequential treatment is first demonstrated in vitro on MIA PaCa-2 cells and finally in vivo on Balb/c mice. Additionally, another important finding is that generation of heat shock protein (HSP) is much reduced in the sequential treatment as opposed to the simultaneous protocol, meaning that lower dosage of chemodrug and photothermal agent can be used, reducing side effects.

All data are reported with standard deviations and satisfying statistics is provided for the in vivo experiments (5 mice for each of the cohorts). Other experimental procedures are described with necessary details, in particular the morphology of the nanocomposites.

This contribution represents an important advance in the field of cancer therapy and will have a broad impact in both the biomedical and photophysics communities. Conclusions are convincing and at this stage, further experiments are not required, as the principle is well established. It will of course have to be checked for each specific case (cancer type, nature of chemodrug and photothermal agent) but this is clearly out of the scope of such a communication.

As a conclusion, I recommend publication after some polishing of the manuscript (and, also some clarifications).

Response: We greatly appreciate the carefully reading and the positive comment from the reviewer. The corresponding polishing and clarifications are listed below as requirement by the reviewer's further comments.

Comment 1. *For instance the introduction is somewhat long with repetitions, the role of*

the two wavelengths used (980 for T measurements and 730 nm for therapy, with low – CT – and high – PTT – power densities) should be described in a clearer way.

Response: We greatly appreciate this valuable advice from the reviewer. The introduction part in the revised manuscript was amended to reduce repetitive descriptions. Meanwhile, in the second paragraph of introduction, we clarified the roles of 980 nm and 730 nm lasers and supplemented the detailed process of programmed combinational therapy. Revised parts of the introduction section are labeled in yellow in the manuscript.

Comment 2. Caption to Fig. 3. Please give Er(III) concentration and, also, excitation wavelength

Response: Thanks for this helpful advice. We indicated the concentration of Er(III) ($2.5 \times 10^{-5} \text{ mol L}^{-1}$) and the excitation wavelength (980 nm) for luminescence spectra comparisons in Fig. 3a.

Figure 3. | Optical and temperature sensing properties of TR-UCNS. (a) Upconversion luminescence spectra of YSUCNP, YSUCNP-PdPc and YSUCNP-PdPc@DPPC-DOX (TR-UCNS) in the aqueous solution with the same concentration of Er^{3+} ($2.5 \times 10^{-5} \text{ mol l}^{-1}$). The excitation wavelength of upconversion luminescence is 980 nm. (b) Absorption spectrum of TR-UCNS. (c) A plot of $\ln(I_{525}/I_{545})$ versus $1/T$ to

calibrate the thermometric scale for TR-UCNS. I_{525} and I_{545} indicate the intensities of UCL emission of the ${}^2\text{H}_{11/2} \rightarrow {}^4\text{I}_{15/2}$ and ${}^4\text{S}_{3/2} \rightarrow {}^4\text{I}_{15/2}$ transitions, respectively. (d) Elevation of apparent temperature (A.T.) and eigen temperature (E.T.) of TR-UCNS (0.5 mg ml^{-1}) in aqueous dispersion under irradiation with a 730 nm laser at various power densities. Average value of A.T. and E.T. under different time points were given based on three times measurements. Error bars were defined as s.d.

Comment 3. There is probably an error in reporting power densities used for the in vitro study (46 and 140 W/cm²) in the text (lines 207, 209), while corresponding captions to figures mention mW.

Response: Thanks very much for pointing out this error. The power densities of 730 nm laser used for in vitro programmed combination therapy are 46 mW cm^{-2} (for the release of drug) and 140 mW cm^{-2} (for photothermal therapy). We corrected the mistaken unit of “W/cm²” into “mW cm⁻²” in the section: “**Results: Programmed combination therapy in vitro.**”. We also checked rest parts of the manuscript to make sure that all units are correct.

Comment 4. There is some confusion in the description of the in vitro experiments. While at the beginning (lines 229-230) is it clearly stated that all cell experiments were conducted with MIA PaCa-2 cells, Figs. 5c/d report images of HeLa cells (as said in the caption). If different cell lines were used, please explain why and, also, clearly state this in the captions to Fig. 5.

Response: We deeply apologize that the “HeLa cells” is a written mistake made in the caption of Fig. 5c. We corrected the mistaken word “HeLa cells” into “MIA PaCa-2 cells”. The cell line used throughout the in vitro study was MIA PaCa-2 human pancreatic adenocarcinoma cell. No HeLa cell was used in this work. We apologize again for the confusion and inconvenience caused by this error. The corrected caption of Fig. 5c is shown below,

“c, Temperature imaging and luminescence imaging (luminescence of TR-UCNS and DOX) of TR-UCNS labelled cells for photothermal triggered chemotherapy with 730 nm laser irradiation (“730 (-)” and “730 (+)” respectively indicate the images before and after 730 nm laser irradiation.). The power density of 730 nm laser used for drug release was 46 mW cm^{-2} . Temperature mapping of MIA PaCa-2 cells were acquired according to the

thermal equilibrium: $(I_{545})/(I_{525}) = C \exp(-\Delta E/kT)$ where I_{545} and I_{525} were the UCL emission intensities in the wavelength region of 535-580 nm and 515-535 nm, respectively.”

Comment 5. *Caption to Figure 5. Please explain the meaning of 730 nm (-) and 730 nm (+)*

Response: We appreciate this useful suggestion. The descriptions about “730 nm (-)” and “730 nm (+)” in Fig. 5c and d were supplemented in the caption of Fig. 5. “730 nm (-)” and “730 nm (+)” are used to indicate the images taken before and after 730 nm laser irradiation.

The revised captions of Fig. 5c and d are shown below,

c, Temperature imaging and luminescence imaging (luminescence of TR-UCNS and DOX) of TR-UCNS labelled cells for photothermal triggered chemotherapy with 730 nm laser irradiation (“730 (-)” and “730 (+)” respectively indicate the images taken before and after 730 nm laser irradiation.). The power density of 730 nm laser used for drug release was 46 mW cm^{-2} . Temperature mapping of MIA PaCa-2 cells were acquired according to the thermal equilibrium: $(I_{545})/(I_{525}) = C \exp(-\Delta E/kT)$ where I_{545} and I_{525} were the UCL emission intensities in the wavelength region of 535-580 nm and 515-535 nm, respectively. **d,** Temperature imaging and luminescence imaging (luminescence of TR-UCNS and PI) of TR-UCNS labelled cells during photothermal therapy with 730 nm laser irradiation (“730 (-)” and “730 (+)” respectively indicate the images taken before and after 730 nm laser irradiation.). The power density of 730 nm laser used for drug release was 140 mW cm^{-2} .”

Comment 6. *Figure 7c. Please give the dosages of drug and photothermal agent used*

Response: We appreciate this valuable advice and apologize for the insufficient explanations of treatment dosages. For evaluating the therapeutic effect of programmed combination therapy, the dosages of chemodrug (doxorubicin) used is $2.5 \mu\text{M}$ and photothermal agent (PdPc) used is $15.3 \mu\text{g mL}^{-1}$.

According to the data in Supplementary Figure 3d and 3e, 0.77 mg PdPc ($2 \text{ mg} \times 76.4\% \times 0.5$) is loaded in 10 mg YSUCNP and $0.115 \mu\text{mol doxorubicin}$ ($2.5 \text{ mM} \times 50$

$\mu\text{l}\times 91.8\%$) is loaded on 10 mg YSUCNP-PdPc (10 mg YSUCNP-PdPc has 0.19 nmol nanoparticles, also say 9.3 mg YSUCNP). For the convenience of quantification and simple representation, the dosage of TR-UCNS in this work are represented by the mass concentration of YSUCNP contained in TR-UCNS (Using mass concentration of YSUCNP to represent the dosage of TR-UCNS can easily quantify the dosages of PdPc and doxorubicin applied in therapy without considering the quantity of DPPC, which is difficult to be characterized by TGA or absorption. The related explanations are supplemented in the manuscript.). Hence, TR-UCNS, of which the mass concentration of YSUCNP is $200\ \mu\text{g}\ \text{mL}^{-1}$, used for in vitro programmed combination therapy study contain $2.5\ \mu\text{M}\ \text{DOX}$ ($0.115\ \mu\text{mol}\times 200\ \mu\text{g}\ \text{mL}^{-1}/9.3\ \text{mg}$) and $15.3\ \mu\text{g}\ \text{mL}^{-1}\ \text{PdPc}$ ($0.77\ \text{mg}\times 200\ \mu\text{g}\ \text{mL}^{-1}/10\ \text{mg}$).

Comment 7. Line 352. Can the authors quantify what they mean by “less dosages”? Which proportion? Getting this information would strengthen the article.

Response: We appreciate this valuable advice. The description of “less dosages” is clarified in **Discussion** section of the revised manuscript. “Less dosages” is proposed based on the comparison of therapeutic effect between conventional combination therapy and programmed combination therapy. If dosages of drug (doxorubicin) and heat (represent by 730 nm laser power density) were kept at low level as used in programmed combination therapy ($2.5\ \mu\text{M}$ of doxorubicin and $\sim 150\ \text{mW}\ \text{cm}^{-2}$ of 730 nm laser), conventional combination therapy that initiate chemotherapy and photothermal therapy at the same time only killed 48.7% of the cancer cells (Figure 6b, column labeled with ‘H’). However, programmed combination therapy achieved a more complete killing effect with only 1.3 % cancer cells survived. This indicates that programmed combination therapy uses less drug and heat to realize an ideal killing effect that conventional combination therapy may reach by using more drug and heat. If the same therapeutic effect as programmed combination therapy, up to 8 folds of chemodrug is needed ($20\ \mu\text{M}$ of doxorubicin) or up to 2.6 folds of 730 nm laser power density is needed ($400\ \text{mW}\ \text{cm}^{-2}$). The supplemented explanation in **Discussion** section is shown below and is labeled in yellow in revised manuscript,

“When the dosages of chemodrug and heat are kept at low level ($2.5\ \mu\text{M}$ of DOX and

heat generated by $\sim 150 \text{ mW cm}^{-2}$ of 730 nm laser), programmed combination therapy can achieve 39 folds improvement in therapeutic effect in vitro than conventional combination therapy that initiates chemotherapy and photothermal therapy at the same time. This indicates that programmed combination therapy uses less drug and heat to realize an ideal killing effect that conventional combination therapy may reach by using more drug and heat. It is worth noting that if the same therapeutic effect as programmed combination therapy is wanted, up to 8 folds of DOX is needed ($20 \text{ }\mu\text{M}$) or up to 2.6 folds of 730 nm laser power density is needed (400 mW cm^{-2}) (Figure S7a and S7b).”

Comment 8. Some suggested text improvements

- *Line 35, replace “received” with resulted in”*
- *Line 39, replace “nanocomposite” with “a nanocomposite”*
- *Line 42, replace “have” with “has”*
- *Line 85, replace “a n on-doping” with “an undoped”*

J.-C. Bünzli

Response: We appreciate the advices for improving our manuscript. The suggested words for replacement have been added in the manuscript and highlighted in yellow. We are sincerely grateful to the reviewer for the recognition of our work as well as the comments to improve the manuscript.

Reviewers' Comments:

Reviewer #1:

Remarks to the Author:

The authors have substantially improved the quality of their manuscript by performing the requested additional experiments, and added significant new data. They also have addressed the earlier scientific concerns. Thus, this article deserves publication in Nature Communications.

Reviewer #2:

Remarks to the Author:

Except for the point developed below, the authors have adequately answered the concerns and suggestions of the reviewers so the ms can be accepted once the following remark is dealt with:

Answer to Comment 1, reviewer 1. The calculations presented by the authors are not correct.

Firstly, there is a mistake in the numerical calculation itself: the given equation on top of page 2 of the response to reviewers (lines 1-3) corresponds to 0.85×10^{-17} g. Second, the volume of a sphere is $\frac{4}{3}\pi r^3$ not πr^3 . The correct value is then 1.13×10^{-17} g. Therefore 20 mg correspond to 2.94 nmol so that according to me there are 435 PdPc molecules in each particle, not 3120.

Moreover note that if the calculated number of moles of PdPc appears to be correct, 1.28 micromole is obtained by dividing $2 \text{ mg} \times 0.764$ by $1197.8 \text{ g mol}^{-1}$ not by multiplying it. All these calculations should be carefully re-checked.

Point-by-Point Response to Reviewers

Reviewer #1 (Remarks to the Author):

The authors have substantially improved the quality of their manuscript by performing the requested additional experiments, and added significant new data. They also have addressed the earlier scientific concerns. Thus, this article deserves publication in Nature Communications.

Response: We appreciate the reviewer for the positive comments of our work and the constructive advices to improve this manuscript.

Reviewer #2 (Remarks to the Author):

Except for the point developed below, the authors have adequately answered the concerns and suggestions of the reviewers so the ms can be accepted once the following remark is dealt with:

Answer to Comment 1, reviewer 1. The calculations presented by the authors are not correct. Firstly, there is a mistake in the numerical calculation itself: the given equation on top of page 2 of the response to reviewers (lines 1-3) corresponds to 0.85×10^{-17} g. Second, the volume of a sphere is $\frac{4}{3}\pi r^3$ not πr^3 . The correct value is then 1.13×10^{-17} g. Therefore 20 mg correspond to 2.94 nmol so that according to me there are 435 PdPc molecules in each particle, not 3120.

Moreover note that if the calculated number of moles of PdPc appears to be correct, 1.28 micromole is obtained by dividing $2 \text{ mg} \times 0.764$ by 1197.8 gmol^{-1} not by multiplying it. All these calculations should be carefully re-checked.

Response: We greatly appreciate the reviewer for this valuable suggestion and apologize for the miscalculation of the volume of nanoparticles. We corrected the equation and

presented the calculating steps in a clearer way. The calculating processes are shown below.

The densities of NaLuF₄:20%Yb,2%Er, NaLuF₄ and SiO₂ are 6.49, 6.52 and 2.40 g cm⁻³, respectively. The diameters of NaLuF₄:20%Yb,2%Er and NaLuF₄:20%Yb,2%Er@NaLuF₄ nanoparticles are 13.4 and 20.5 nm, so the radii of NaLuF₄:20%Yb,2%Er and NaLuF₄:20%Yb,2%Er@NaLuF₄ nanoparticles are 6.7 and 10.25 nm, equivalent to 6.7×10⁻⁷ and 10.25×10⁻⁷ cm, respectively. The external and internal diameters of yolk-shell silica layer are 48.1 and 36.5 nm, so the radii of the external and internal yolk-shell silica layer are 24.05 and 18.25 nm, equivalent to 24.05×10⁻⁷ and 18.25×10⁻⁷ cm, respectively. The correct weight of one NaLuF₄:20%Yb,2%Er@NaLuF₄@Yolk-Shell SiO₂ (YSUCNP) nanoparticle is 10.8×10⁻¹⁷ g, that is $4\pi/3 \times \{[(24.05 \times 10^{-7} \text{ cm})^3 - (18.25 \times 10^{-7} \text{ cm})^3] \times 2.40 \text{ g cm}^{-3} + [(10.25 \times 10^{-7} \text{ cm})^3 - (6.70 \times 10^{-7} \text{ cm})^3] \times 6.52 \text{ g cm}^{-3} + (6.7 \times 10^{-7} \text{ cm})^3 \times 6.49 \text{ g cm}^{-3}\}$. Hence, the correct molar of 20 mg YSUCNP is 0.31 nmol, which is calculated from (0.02 g/10.8×10⁻¹⁷ g)/6.02×10²³ mol⁻¹. Meanwhile, the number of PdPc molecules combined with 20 mg YSUCNP is 1.28 μmol, that is (2 mg×76.4%)/1197.8 g mol⁻¹, of which the calculation equation is corrected as the comments of the reviewer. As a result, there are ~4129 (calculated by 1.28 μmol/0.31 nmol) PdPc molecules in each YSUCNP nanoparticle. Based on the data in Supplementary Figure 3e, 10 mg YSUCNP@PdPc, which contain 9.3 mg YSUCNP, equivalent to 0.14 nmol ((0.0093 g/10.8×10⁻¹⁷ g)/6.02×10²³ mol⁻¹) YSUCNP nanoparticles, are loaded with 0.115 μmol (2.5 mM×50 μl×91.8%) DOX molecules, so there are ~821 (calculated by 0.115 μmol/0.14 nmol) DOX molecules on each nanoparticle.

We carefully checked other calculations and confirmed that they are correct. The corrections to the numbers of PdPc and DOX molecules do not affect the results of other parts of the manuscript.